# Early Childhood Teachers' Support of Children's Play in Nature-Based Outdoor Spaces—A Systematic Review

Tor Mauritz Smedsrud [1,2,*], Rasmus Kleppe [1,3], Ragnhild Lenes [2] and Thomas Moser [2]

1 Kanvas Foundation, 0179 Oslo, Norway; rasmus.kleppe@dmmh.no
2 Norwegian Centre for Learning Environment and Behavioral Research in Education, University of Stavanger, 4036 Stavanger, Norway; ragnhild.lenes@uis.no (R.L.); thomas.moser@uis.no (T.M.)
3 College of Early Childhood Education, Queen Maud University College, 7044 Trondheim, Norway
* Correspondence: tor.smedsrud@kanvas.no

**Abstract:** Early Childhood Education and Care (ECEC) places value and benefits on children's play in nature-based outdoor spaces. However, there is a lack of knowledge regarding teachers' support of play in environments with rugged terrains and natural materials. Therefore, this systematic review aims to locate, present, and discuss research literature on how teachers in ECEC settings can support children's play in nature-based outdoor spaces. According to the review, teachers' support of children's play was described in the literature as encouraging children's free and unstructured play through facilitating actions addressing the children's perspectives and the opportunities offered by the physical environment. Furthermore, teacher support was also described as teacher-led and teacher-guided interactions where teachers and children communicate or collaborate in playful situations. Differences and overlapping elements of types of teacher support are discussed, and implications for researchers, practitioners, and ECEC teacher education are provided.

**Keywords:** early childhood education; nature-based outdoor spaces; natural environments; play; learning; child development; teacher support





## 1. Introduction

This systematic review aims to locate, present, and discuss research literature on how teachers in Early Childhood Education and Care (ECEC) settings support children's play in nature-based outdoor spaces.

Nature-based outdoor spaces have been regarded as beneficial for children's health, learning, and development; there is empirical support that children who spend time in natural environments develop knowledge about the environment, positive environmental attitudes, and connection to nature [1,2]. Furthermore, proximity to green spaces is associated with increased physical activity and improvements in motor skills [1,2]. Higher levels of physical activity have been documented in natural environments compared to ordinary school environments for children between the ages of three and eighteen [3], as well as a conditional support for benefits on self-esteem, self-efficacy, resilience, and academic and cognitive performance [3]. Improvements in cognitive outcomes, such as play, learning, and creativity have also been shown [4]. In early childhood development, these apply to areas such as attention, punctuality, concentration, constructive play, associative play, imaginative play, and functional play [4]. A diverse natural environment can provide children with opportunities to participate in different types of play that are likely to influence social, emotional, and cognitive capacities [5]. Prins et al. [6] conclude that play in natural environments has high quality because of high levels of socio-dramatic, explorative, and risky play [6]. Overall, a recent review [7] concludes that nature exposure for children is beneficial for their well-being and development. Considering these benefits reported in the literature, it is not surprising that authorities, educators, and researchers internationally take an interest in the use of natural environments in ECEC [8–11]. The growing interest can

also be seen in the context of an increasing focus on the environment and sustainability in ECEC [12]. However, there is great variation in types of natural environments, and nature as a context can be understood in many ways. Therefore, we use the term nature-based outdoor spaces to clarify that the natural environments included in the review are outdoors, with characteristics of nature-based elements, such as rugged terrain and natural materials. When referring to the literature, we apply the terms used by the authors (e.g., natural environment, green spaces).

Play is an ambiguous concept and can be described and understood in many different ways [13]. Play can be considered as children's intrinsically motivated activities in and of itself, often referred to as free or unstructured play. It can take form as children socializing, using language and exploring the world around them, often in connection with fantasizing imaginary role play situations [14]. It has been suggested that unstructured play improves children's physical, emotional, and social well-being [15]. Nevertheless, there is a debate in ECEC to what extent teachers should involve themselves in children's play [16], as well as some consensus that teachers should contribute to playful activities leading to learning and developmental outcomes [17]. Therefore, the term play is also used in contexts where the activities are teacher-led or teacher-guided. Research suggests that teachers' involvement in play positively influences children's cooperativeness [18,19]. Moreover, recent studies have shown that children in ECEC settings want teachers to be present in the play to meet their emotional needs (for example, in the context of peer interactions of exclusion and rejection [20]). Prins et al. [6] found that the role of the teacher influences play quality in nature-based environments. Children's play quality improved when they were given more independent mobility licenses [21] from their teachers, meaning that teachers transferred some of their nature-based affinity and initiative to the children simultaneously as they had the freedom to play without too much adult supervision [6].

Teachers' support of play in nature-based outdoor spaces is characterized by improvisation and understanding of risk, and a child-centered approach to play and experiences that depend on what the natural environment conveys [22–24]. In ECEC, teachers and children interplay with the physical environment, and the physical environment plays back [25]. It can be assumed that different types of environments contribute to different practices. Nevertheless, previous research has indicated that early childhood teachers often have a limited understanding of the educational potential of children's encounters with natural environments [26–30]. Understanding the educational potential is a prerequisite for teachers to be able to implement supportive learning and developmental measures in this play environment [31,32]. The theory of children's proximal development [33] emphasizes how a core task of the caregiver is to recognize what the child can manage with just the right amount of help (not too much, not too little). The ability to provide this type of appropriate support is termed scaffolding [34]. We consider that examining the relationship between the educational potential of natural environments and the teacher's supportive actions, such as scaffolding, in play will contribute to better understanding of the role of the teacher when practicing in nature-based outdoor spaces.

Based on the growing interest in the role of ECEC staff in terms of instructional support [35] and the beneficial outcomes of play in nature-based outdoor spaces for children's well-being and development, we want to answer the following research question, which we consider to be of relevance for researchers, practitioners, and ECEC teacher educators:

What is known from the literature about how teachers in ECEC settings support children's play in nature-based outdoor spaces?

## 2. Methods

This systematic review followed a systematic, rigorous method [36], using the Preferred Reporting Items for Systematic Reviews and the Meta-Analysis (PRISMA) guidelines [37]. A PRISMA checklist is provided in Supplementary Materials (File S1). Based on the research question, the following inclusion criteria were defined:

Type of publication: Empirical peer-reviewed papers published in academic journals and book chapters in peer-reviewed anthologies.

Time of publication: 2002–2023. We were interested in recent research literature and therefore decided to set the starting point for our search to the beginning of the 21st century. The search process was finalized on 28 May 2023, and all relevant publications identified until this date were included.

Language: Papers written in English, Norwegian, Danish, or Swedish.

Participants: Teachers and children in ECEC settings. We define teachers as all staff and practitioners working in physical environments where children, about one to six years of age, stay for a time-limited period of the day. We exclude elementary school teachers and children, parents and children, and leisure time activities.

Context: Physical places outdoors where there are rugged terrain and natural materials. We exclude physical places outdoors with fewer characteristics of nature-based elements, e.g., playgrounds, ECEC settings' outdoor spaces, parks, and museums.

Concept: The paper must contain descriptions of teachers' support of children's play in nature-based outdoor spaces. Teachers' support can take the form of actions that extend the goal of play (in and of itself) and play as an instrument for children's learning and development. For papers to be included, they needed to incorporate the concept of play or offer explicit descriptions of the processes and activities that define play behavior. Teacher support includes all descriptions of teachers' planning and facilitation of play and their interaction with children in play situations.

*2.1. Screening Process*

An initial search in Academic Search Premier and ERIC databases identified synonyms for participants, context, and concept. Synonyms to the ECEC settings were found in the Nordic Base of Early Childhood Education Research (NB-ECEC) database [38]. Additional synonyms were found in cooperation between the authors. A second search identified keywords and index terms. The search string is presented in Supplementary Materials (File S2). A final search was conducted in Academic Search Premier, ERIC, Web of Science, and ORIA (Norwegian professional library) databases. Based on the experiences from earlier literature reviews, e.g., the NB-ECEC [38], a search of these databases will provide us with a sufficient proportion of relevant studies. Three additional papers were found by hand searching reference lists of received papers.

Hits from the search process were imported to EPPI-Reviewer 6 software [39]. Duplicates were removed ($n$ = 391), and the first author screened titles and abstracts to remove papers that did not meet the inclusion criteria. To reduce the risk of potential bias, ten percent ($n$ = 136) of the remaining 1361 publications were randomly selected and independently double-screened by two reviewers. No instances of disagreement were identified. A total of 1257 papers were excluded, and 104 papers were retrieved for full-text screening. The remaining 104 full-text publications were screened independently by two authors, and in cases of disagreement between the authors, a third author made the final decision.

Some papers seemingly met all inclusion criteria but were excluded. For example, on the criterion 'type of publication', one paper [40] was excluded because no method section was provided, although the paper could be regarded as qualitative research with thick descriptions. Regarding the criterion 'context', one paper [27] was excluded because it depicted physical places outdoors with fewer characteristics of nature-based elements. Considering the criterion 'concept', several papers were excluded because of not incorporating the teacher's role, e.g., one paper [41] described the impact of nature encounters and children and nature relations but did not incorporate teacher support. All included papers are presented in Supplementary Materials (File S3) in alphabetical order. In one of the papers that are included [42], there are portrayals of three contexts, of which one meets the inclusion criteria. In this case, we have disregarded the parts portraying the wrong context (courtyard space) and the wrong participant (childminder). A PRISMA flow diagram [43] is provided in Figure 1.

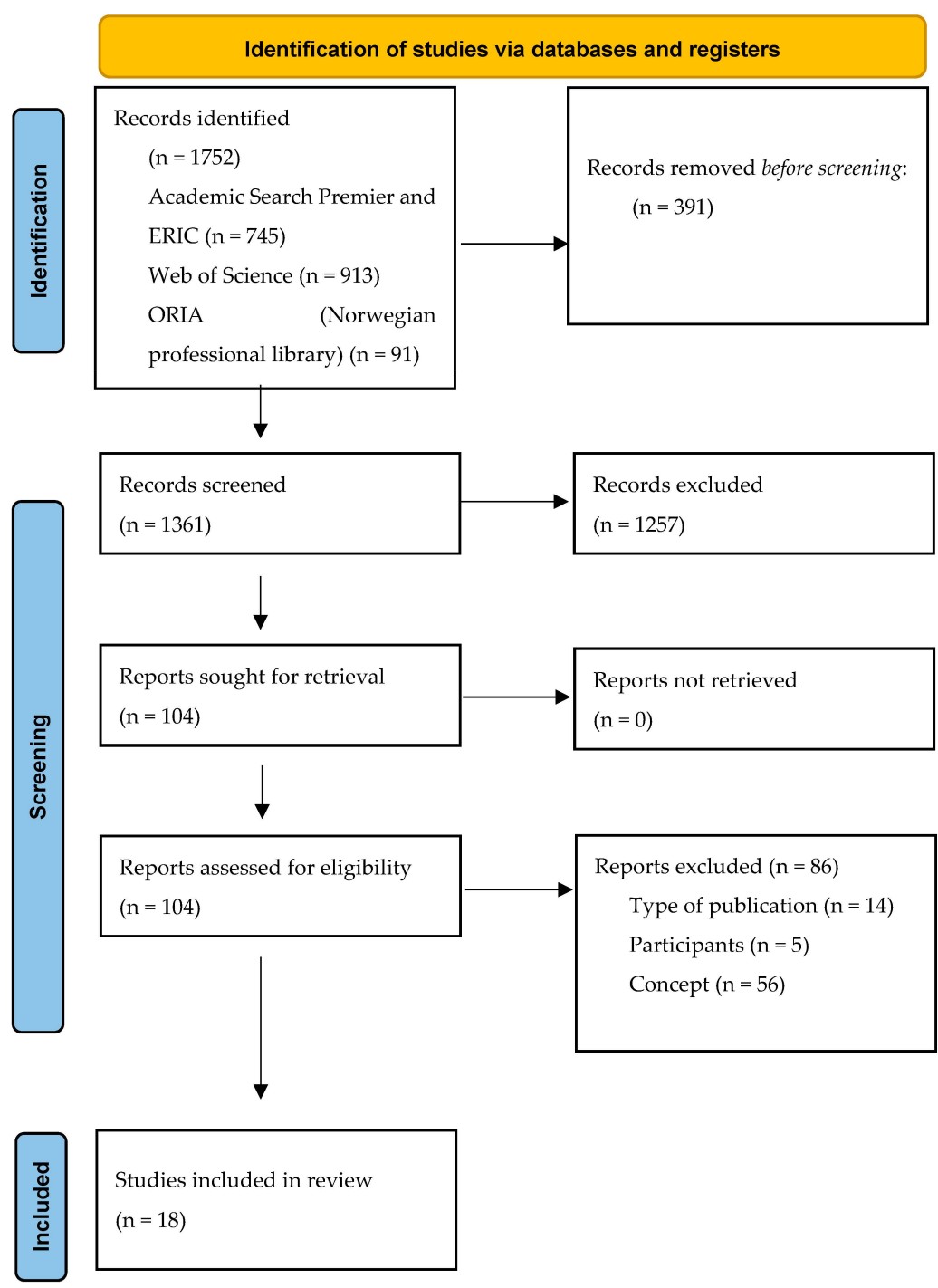

**Figure 1.** PRISMA flow diagram.

*2.2. Data Extraction and Synthesis*

The authors collaborated to make codes to extract relevant information from the papers. The codes were (1) location, (2) participants, (3) context, (4) aims/research questions, (5) concept of children's play (theoretical framework), (6) concept of teachers' support (theoretical framework), (7) conclusions, (8) descriptions of how teachers interact with the children and the natural environment, and (9) descriptions of the use of material objects and the inherent properties of rugged terrains and natural materials. Suitably, all authors coded three of the eighteen included papers to evaluate whether we assessed the codes similarly. When necessary, we discussed calibrating our understanding of the codes. The first author extracted data from the included papers using the EPPI-Reviewer 6 software [39]. Next,

the data were extracted from the coding reports into a more compact and readable table (see Supplementary Materials, File S4). The co-authors read their assigned papers (six for each co-author). No major discrepancies were detected. The authors categorized and synthesized the extracted data and applied core elements of Noblit and Hare's [44] meta-ethnography method, i.e., repeated reading of the papers and multiple meeting points between the authors to discuss emerging findings and the meanings of key concepts. The authors collaborated in examining key concepts, such as 'play' and 'teachers' support of play', within and across the papers. The final disposition of the synthesis results is based on a dialogical process between the authors.

The included papers consisted exclusively of qualitative studies. It has been debated how and whether it is relevant to appraise the quality of qualitative research in systematic reviews [45]. We found it relevant for this review to report our perception of the trust-worthiness of the findings, answering a question derived from the JBI checklist [46]: 'Are the objectives, methodological approach, data collection, and data analysis clearly stated, and is there congruity between them?'. Thus, our quality assessment mainly indicates the reporting of methodological issues. Two authors independently assessed the trust-worthiness of the included studies to be either high, medium, or low. Our assessments of trustworthiness, with reasons, are presented in Supplementary Materials (File S4).

## 3. Results

The results are expressed through multiple layers of contexts to enable the reader to understand the descriptions of teacher support in relation to locations, participants, characteristics of the environments, aims, methods, and the concept of play and teacher support. Thus, the findings are presented in four sections, reporting (1) locations, participants, and characteristics of the environments, (2) aims and methods, (3) the core concepts of play and teachers' support of play, and (4) how teachers support children's play in nature-based outdoor spaces. The terminology used in the original paper is preserved in the synthesis. Regarding the quality assessment, all articles were rated medium or high, thus it can be assumed that the trustworthiness of the studies is acceptable or good.

### 3.1. Locations, Participants, and Characteristics of the Environments

The 18 included papers were from seven different countries. Of 25 included ECEC settings, 21 referred to a nature-based provision profile (see Table 1).

**Table 1.** Overview of locations and specified types of ECEC settings.

| Location and Number of Papers | Types and Number of Included ECEC Settings | Context |
|---|---|---|
| Australia (*n* = 4) | Bush kinders (*n* = 4) | An Australian approach to nature-based ECEC, with roots in Scandinavia and Nordic countries [9] |
| United Kingdom (*n* = 4) | Outdoor and forest kindergartens in Scotland (*n* = 5)<br>Forest school sessions of a nursery class in England (*n* = 1)<br>Private day nursery with its woodland on the borders between Wales and England (*n* = 1) | Forest school is strongly associated with the Danish ECEC, emphasizing play, movement, and fresh air. Numerous projects are being established across the country [47] |
| Norway (*n* = 3) | Nature kindergartens (*n* = 7)<br>Kindergarten outdoor group (*n* = 1) | ECEC settings that are often outdoors and frequently use specific natural spaces outside their fences [48] |
| Sweden (*n* = 2) | Outdoor preschools (*n* = 2) | Members of the Swedish Outdoor Association, with a focus on nature and the environment [11] |
| Canada (*n* = 2) | Forest school (*n* = 1)<br>Forest school practitioners | Inspired by similar programs in northern Europe [24] |

| Location and Number of Papers | Types and Number of Included ECEC Settings | Context |
|---|---|---|
| New Zealand (*n* = 2) | Early childhood education centers (*n* = 2) | ECEC settings that organize outings to protected bush reserves and weekly visits to the 'wild woods' [49,50] |
| Denmark (*n* = 1) | Forest kindergarten (*n* = 1) | ECEC settings that are often outdoors and frequently use forest sites [51] |

The descriptions of the nature-based outdoor spaces varied, but they all had features of rugged terrain and natural materials, mostly related to places in woodlands and forests, within walking distance from the ECEC settings, and with a lack of fences. Features of the woodland and forest environments included native pinewoods and broad-leaf forests [52], snow, mud, plants, stones [53], sticks, branches, leaves [42,54], grass, rock formations, and fire pits [50]. Sometimes, there was still or running water [50,55,56]. In one study [51], the authors classified forest preschool sites with ten outdoor features: open ground, sloping terrain, shielded places, rigid fixtures, moving fixtures, loose objects, loose material, water, creatures, and fire. Although one paper [57] defined natural environments as not created by humans, the distinction between natural and cultural landscapes was rarely clarified [50,51,58]. Play materials were mainly self-made, and adapted to the natural environment, such as swings in trees made from ropes [56,58]. Only a few pre-defined pieces of play equipment and commercial toys were available for the children. Some papers included seasons and weather conditions as relevant factors in using nature-based outdoor spaces [50–53]. In some papers, cultural features were attributed to the descriptions, e.g., relating natural environments to Nordic traditions and values [56,57,59], and cultural heritage in relation to indigenous Māori protocols [49]. Trips to and sessions in the forest and woodland appeared to be regular. The described groups accommodated between 12 and 25 children and three to six teachers [51,53,56–64]. The studies included one to six ECEC settings, one to one hundred children aged one to eight years, and one to twenty teachers. In some papers, the number of children or teachers was not specified. The teachers who are portrayed in the studies were described in different ways, e.g., as staff, pedagogues, practitioners, teachers, educators, forest school leaders, adults, and assistants. The educational level of the teachers ranged from no education to higher education, and they had different levels of experience.

Most papers did not define nature-based outdoor spaces explicitly but described the natural environment and referred to the benefits of engagement. The terms used in the papers were, nature, natural environments, outdoor environments, natural surroundings, outdoor natural settings, outdoor spaces, forest sites, and green settings. Lerstrup and Refshauge [51] defined forest sites as "sites in public accessible green settings used for preschool stays, often named" and green settings as "outdoor settings where forest features are perceived as dominant; including natural, semi-natural and cultivated areas" (p. 388). Some papers related the physical environment to a theoretical framework, such as a 'socio-cultural context' [53], 'affordances' [50–52], and 'theory of loose parts' [50,63].

*3.2. Aims and Methods*

The papers had many aims and research questions; only two had the support of children's play as a primary goal [42,54]. Some papers examined developmental and learning outcomes, such as children's involvement [60], autonomy [52], identity [57,63], and science learning [61,62]. Other papers examined educational practices, such as teacher's didactics [53], communication [59], democratic practices [56], and pedagogical approaches [64]. Two papers explored perspectives on risky play [55,65]. Other objectives included characteristics of forest sites [51], different ways in which children and teachers perceive and utilize the affordances offered by the environment [50], children's participation [58], and practical procedures of observable moral work [49]. The four papers from Australia were based on a

longitudinal project titled 'Bush kinders: Locating the Science', depicting bush kinders from the same shire in the country's southeastern part [61–64]. Two of the papers from England were built on a rural private day nursery research project with its woodland [42,54].

Eleven of the papers explicitly framed the study within ethnography and used terms such as ethnography [62,64], ethnographic approach [57,63], ethnographic methodology [52], small-scale ethnographic study [54], ethnographic design [61], sensory ethnography [53], ethnomethodological framework [49], ethnography-inspired method [51], and interpretivist ethnographic case study [50]. Data collection in these studies varied, with field observations, photographs, video and audio recordings, and conversations and interviews as main features. The data collection of the papers that did not explicitly frame the study within ethnography was based on focus group interviews [58], non-participant observations [42], observations and semi-structured interviews [60,65], video observations [59], and field notes, video recordings, and interviews [56]. One study conducted semi-structured interviews with teachers completed by phone [55]. The ways data have been analyzed, findings discussed, and conclusions drawn varied considerably, partly depending on the aim of the study, the design, and data collection. Our general impression is that the papers gave detailed descriptions of children's play in natural environments and provided educational suggestions for outdoor ECEC; however, the support of children's play was not a central theme that was directly addressed, but appears in a rather indirect way.

### 3.3. Understanding the Core Concepts of Play and Teachers' Support of Play

The core concepts related to the research question of this review are play and teachers' support of play. The term play occurred in all included papers. Yet, there were variations regarding the context in which the term appeared. Some papers understood play as children's self-governing activities, such as free play [50,51,57,58], and unstructured play [42,62]. Other papers understood play as social and symbolic and included terms such as fantasy play [58], pretend play [49,57], imaginative play [42], and creative play [54]. Some papers used the term closely related to adult–child communication and learning experiences and applied terms such as playful reasoning [59] and play-based learning [61,64]. Other papers looked at play in relation to risk [55,65] and gender [63]. Often, it seems like there are fluid transitions between the perspectives mentioned here. Most papers did not explicitly define the concept of play. However, Sanderud et al. [53] referred to self-governing play as "activities that are non-compulsory, driven by intrinsic motivation and undertaken for [their] own sake, rather than as a means to an end" (p. 1089). Harper and Obee [55] stated, "risky play has been defined as thrilling and challenging forms of play with the potential for physical harm" (p. 185). Speldewinde and Campbell [63] defined nature play as "[...] children's play that is specifically unstructured and nature-based" (p. 817). Three papers did not refer to any understanding of the play. These papers [51,57,59] still had detailed descriptions of processes and activities characterizing a kind of play behavior.

The papers generally did not provide explicit conceptualizations of teachers' support of play. However, they analyzed the understanding of teacher support through various theoretical lenses. Possibly because of large variations in objectives and research questions, there seems to be little consistency between the papers regarding the type of support. For example, in one paper [58], the authors used 'a spatial and relational approach to participation' (p. 116) to analyze what we interpret as teacher support. Another paper analyzed active support of the children's tendency to lead themselves through 'self-determination theory' [52]. Other theoretical lenses applied in the papers for what we would regard as analysis of teacher support include: 'didactic sensitivity' [53], 'variation theory' [59], 'well-being and involvement' [60], 'social constructivist epistemology approach' [62], 'hermeneutics of empathy' [55], 'interaction on a turn-by-turn basis' [49], and 'notion of play-based learning' along with a model of 'inquiry-based teaching' [64]. Three papers [50,51,57] used the theory of 'affordances' to relate teacher practices to what the natural environment provides children and teachers. Four papers use a variety of theoretical perspectives in their analysis and

provide descriptions of teacher support related to den-making play [42,54], risky play [65], STEM (Science, Technology, Engineering, and Mathematics) play [61], and democracy and inclusion [56].

### 3.4. Descriptions of Teachers' Support of Children's Play in Nature-Based Outdoor Spaces

The main feature of teachers' support of play was described as encouraging children's free and unstructured play through facilitating actions addressing children's perspectives and the opportunities offered by the physical environment. Furthermore, teacher support was also described as teacher-led and teacher-guided interactions where teachers and children communicated or collaborated in playful situations. However, it is prominent in several of the papers [42,53,57,64] that encouragement, facilitation, communication, and collaboration arise together and overlap each other. Nevertheless, there are differences that are evident in terms of the descriptions of the type of play and the type of teacher support (see Table 2).

**Table 2.** Overview of type of play and type of teacher support.

| Type of Play and Number of Papers Where It Is Described | Type of Teacher Support |
|---|---|
| Free and unstructured play (*n* = 14) | Mainly encouraging play options for the children. Teacher role largely either absent or observant, a few descriptions of teacher's participation in free play |
| Teacher-led and teacher-guided play (*n* = 9) | Communicative and collaborative interactions between teachers and children. Teachers motivating, offering advice, explaining, clarifying, and challenging the children |

As a core facilitating element, the site choice was determined by the location of the ECEC settings, the teachers, and, in some cases, through inputs from the children [53]. According to Lerstrup and Refshauge [51], the choice of site was based on the range of features on the sites and how the site was situated and on such "[...] factors as knowledge about the abilities of the actual group of children present, the need for surveillance, and the level of staffing" (p. 395). The included papers indicated that children were being introduced to the chosen site. Teachers and children were primarily exposed to forest environments with rugged terrain and natural materials. These environments provided open-ended play opportunities, although teachers adjusted or modified the environment [53], removed hazards [65], and prioritized certain activities [42,56]. Teachers encouraged various play options by rousing the children's wonder and curiosity [53,54,59] and showing how the environment could be used symbolically in pretend play [49,57]. Several of the included papers highlighted that the children were allowed to choose their activities [50,52,53,57,58]. Children's play was mainly undisturbed by the influence of teachers, and the teacher's role was largely either absent or observant [42,50,52–54,56,58,65]. This concoction of teachers' encouragement and non-disruptive behavior was particularly emphasized in the two papers that explicitly aim to explore teachers' support of play [42,54]. One of these two papers [54] portrayed a casual comment by a teacher about 'bears in the woods'. The comment became the focus of conversation between the children, and they created a narrative about a family of bears living in the woods. The narrative was further developed by the children and facilitated by the teachers. The narrative created options for children's imaginative and creative play. Canning [54] observed that "[...] the practitioners placed significance on the natural environment as a resource for learning and stood back from the children's play to enable them to follow their own interests" (p. 1051). In the other of the two papers [42], it was emphasized that teachers were sensitive to their own influence on children's experiences and exploration of own interests. Following the children's interests was a recurring theme in several papers [42,52,54,58,60,64]. It also emerges that the

children, to some extent, were made responsible for their actions: for example, through expectations of managing risk and other situations [53,65], as well as to approach teachers if they needed assistance [53]. Moreover, some of the papers suggest that the traditional child–adult dichotomy was challenged. For example, it is mentioned that sites were contextualized by both teachers and children [56], pretend play offered possibilities for children to re-formulate adult rules [49], enactment of teacher restraint gave children responsibilities that required collaboration, communication, and creativity [58], and teachers were learning from the children [55].

In total, 14 papers [42,49–58,60,64,65] stated that teachers were crucial in encouraging and facilitating children's free and unstructured play, sometimes referred to as social and symbolic play. It was reported that teachers refrained from regulating [58] and rarely participated in play [56]. However, there were contradictions in the findings. It was also described that teachers participated in free play [57], took initiative for 'rule games' [56], and conducted fantasy play together with the children [58]. In nine papers [49,50,57,59–64], there were descriptions of more teacher-led or teacher-guided play interactions. In these interactions, teachers provided positive experiences and playful interactions: motivating, offering advice, explaining, clarifying, challenging, and extending children's learning and development [52,59,61,63]. These interactions were portrayed as both child-centered and teacher-initiated playful learning situations. Mawson [50] stated that [. . .] "the children's day-to-day experiences were dependent on the particular approach being used by the teachers who had accompanied them to the woods" (p. 518). One study [64] found that different bush kinders had varying pedagogical approaches, while the findings from another study [60] indicate that a nursery school teacher and a forest school leader had distinctly different ways of planning and implementing teacher support. It was observed that the nursery school teacher had flexible planning and teacher-directed implementation, while the forest school leader had clear routines and an implementation allowing for children's experimentation and responsibility. Mackinder [60] suggested that this "indicates that planning can be closely connected to the training of the adult and the implementation of Forest School" (p. 188). Altogether, the results provide partially overlapping descriptions of teachers who facilitate external conditions for children's free and unstructured play and teachers who interact intentionally in the children's play, often with a learning objective.

## 4. Discussion

The discussion of what is known from the literature about how teachers in ECEC settings support children's play in nature-based outdoor spaces is presented in the same order as the results and relates to the theory in the introduction. The theory will be used to highlight the relationship between the educational potential of natural environments [1,2,4,6,25,31,32] and the teacher's supportive actions in play [6,17,18,20,34,66].

### 4.1. Locations, Participants, and Characteristics of the Environments

In the results, we found that 21 out of 25 included ECEC settings had some kind of nature-based profile. ECEC settings with nature-based foundations can be traced back to the educational idea of 'kindergarten' in the early 19th century [67]. Explicit nature-based programs appeared later, especially in the Nordic countries in the 1980s [11], and were inspired by Danish forest preschools in the United Kingdom in the 1990s [68]. The nature-based provision spread to Canada and Australia in the 2000s [9,24]. Unsurprisingly, most of the included studies were conducted in these countries and mainly in nature-based types of ECEC settings, both because this type of provision is established in these countries and because of the included languages of this review. In the screening process, we came across some papers written in English, e.g., from Turkey [69,70], the Czech Republic [71], and Slovenia [72], but the vast majority of the studies were conducted in Nordic countries or the Anglosphere (United Kingdom, Ireland, Canada, New Zealand, and Australia).

The included papers' descriptions of nature-based outdoor spaces were mostly related to places in woodlands and forests. It can be assumed that children's play and teacher

support of play depends on the possible interplay with the physical environment [25]. Although not thoroughly explained in the papers, it can be speculated that forest play features, such as climbing, hiding, and shelter will influence how teachers support children's play. The same applies to different weather and seasonal conditions. For example, a snowy winter landscape in Norway will probably provide different practices than a warm beach session in Australia. Our general impression is that the papers referred to the benefits of engagement with natural environments, and, to some extent, they relate to value-based attitudes, especially to what often is referred to as Nordic traditions and values. The fact that all the included papers are from the Nordic countries and the Anglosphere may have implications for the understanding of the inherent properties of natural materials and rugged terrains, as well as play in natural environments and how teachers' support children's play. The lack of conflicting views may contribute to underpinning a Euro-Western, and particularly Nordic, understanding of nature-based approaches in ECEC [73]. It may possibly leave out alternative teacher actions in describing teacher support of play in nature-based outdoor spaces. For example, it has been suggested that teachers and parents from southern European countries are more risk aversive to children's play outdoors than teachers and parents from Norway [74]. Furthermore, we found it interesting that the possible negative aspects of children's play in nature-based outdoor spaces were almost absent in the included studies, although Gill [1] emphasizes that negative experiences in nature can contribute to the development of fear, discomfort, and aversion to the natural environment.

*4.2. Aims and Methods*

The included papers were heterogeneous regarding aims and research questions. The fact that only two papers had the support of children's play as a primary goal and that our objective appears indirectly in the other publications may question the validity of the results. Nevertheless, the heterogeneity may also reflect a great variation in interests, theories, and philosophies in both research and practice, and this perspective can contribute to strengthening the understanding of the results. The included papers were relatively homogeneous in terms of participants and data collection. The varied use of ethnography in the included papers may reflect a need to gather direct information from a social phenomenon not well explored in the research literature. It may also be that ethnography turns out to be an appropriate method to unveil aspects of rather heterogeneous educational practices. Studies with approaches other than ethnography could highlight other perspectives on teacher support. Different research methods, e.g., interview and observation, can come up with different answers to the research question. One should have in mind that what informants say in interviews does not necessarily correspond to what they do in practice [75]. Furthermore, perspectives from other places than the Nordics and the Anglosphere could possibly provide further insight into teacher support. More variety in methods and locations could contribute to a better basis for generalizing the descriptions of teacher support in this review.

In educational research [76], it has been suggested that there are some differences between the Nordic countries (continental) and the Anglosphere regarding disciplines and theory. In Nordic countries (continental tradition), education is largely seen as a field in its own right, with concepts such as pedagogy and didactics used to study the objects of Erziehung and Bildung, while English-speaking countries have a clearer orientation towards a disciplinary approach to educational studies [76]. In our findings we could not identify a clear distinction between the Nordic and Anglosphere publications, although one Norwegian paper [53] framed the theoretical work within 'Bildung as playful self-formation' (p. 1086), and three of the Australian contributions highlighted science disciplines [61–63]. Based on our findings, it is difficult to assess disparity between countries in the descriptions of how teachers in ECEC settings support children's play in nature-based outdoor spaces. Publications from both the Nordic countries and the Anglosphere described partially overlapping portrayals of teachers who facilitated external conditions for children's free

and unstructured play, and teachers who interacted intentionally with children in their play. Most of the ECEC settings in the included studies had some kind of nature-based profile, largely building on similar educational traditions and values [68,77]. This can possibly create resemblances between them regarding teacher support. It is also worth noting that recent research has questioned the apparent dichotomy of the Nordic model and the Anglosphere model, suggesting that the dichotomy could be caused by an implicit critique of the authors' own Anglosphere educational context [78]. However, it should be noted that Hindmarch and Boyd [79] reported striking differences in how teachers delivered forest schools in Norway and England, portraying, for example, more rigid planning, set activities, and use of control questions in England compared to Norway (p. 13).

### 4.3. The Core Concepts of Play and Teachers' Support of Play

The included papers were heterogeneous regarding the understanding of play and the use of analytical lenses for examining teacher support. There was a lack of clear conceptualizations of play, and it may seem that the term play in many cases was used somewhat randomly and arbitrarily. However, our impression is that play was understood as important both in and of itself and as an instrument for more teacher-led and teacher-guided learning situations. It is noteworthy that only a few papers discussed nature play as a separate concept, as suggested by Prins et al. [6]. Based on the included papers, it is difficult to differentiate between play in general and play in natural environments in particular. It may be that the results from this review reflect types of teacher support which are also prominent in settings other than nature-based outdoor spaces. Furthermore, we have interpreted descriptions of social and symbolic play as mainly free and unstructured, although teachers might have been more actively involved in these types of play than the impression given by the publications.

Several of the included papers used the terms 'children's interests' describing the concept of play, as well as 'child-centered' in their discussions of teacher support. In ECEC, there is a clear tendency towards more child-centered educational approaches, particularly with the background of the UN Convention on the Rights of the Child [80]. However, how these concepts are related to play and teachers' support of play is not explicitly discussed in the papers. Nevertheless, our reading of the papers suggests that children's free and unstructured play in natural environments contributes to children following their own interests. Furthermore, there are several aspects of nature-based outdoor spaces that spark children's interests, such as, for example, small animals and risky elements. It seems that teachers in many cases built their playful interactions with the children on what the children found interesting in the natural environments.

### 4.4. Descriptions of Teachers' Support of Children's Play in Nature-Based Outdoor Spaces

The continuum of teacher support ranged from descriptions of being absent to elements of dominating and authoritative behavior. Yet, as described in the results, the most prominent type of teacher support of play was teacher's encouragement and facilitation of children's free and unstructured play. In other words, teachers were facilitating the external conditions for play. In our synthesis, this emerges as the choice of site, modifying the environment, removing hazards, and encouragement of various play options. Examples of encouragement and facilitation include showing and telling children alternative play options and providing a few regulations. A degree of trust was given to the children to let them play without adult interference. As we see it, this type of teacher support coincides with the conceptualization of nature play quality developed by Prins et al. [6], as children are given greater mobility licenses from their teachers [21]. It also corresponds to the idea of scaffolding [34], in terms of, for example, choosing sites that potentially give physical challenges, modifying the environment to expand the children's current developmental stage, and providing play options that can create peer-learning opportunities. In several of the included papers, children's self-governing, free, and unstructured play, and social and symbolic play are attributed to beneficial learning and developmental outcomes. This

belief is consistent with research stating that play in natural environments contributes to children's environmental attitudes, attachment to nature, physical development, and emotional well-being [1,2,4] However, in the synthesis, there are only a few examples of teachers explicitly expressing the educational reasons behind the facilitation of external conditions and the reasons why it is appropriate to withdraw from the play situation. For the actions to be called teacher support, we assume that the facilitation needs to be based on the teacher's knowledge of the relationship between the potential of the natural environment and the children's play [31,32]. It is often challenging to conclude whether teacher support has taken place. The somewhat imprecise descriptions of facilitating teacher support may indicate that the teachers' actions result from (tacit) knowledge that is difficult to articulate for teachers and observers. On the other hand, it is possible that many of the descriptions of children playing freely are caused by a few teacher actions, or the actions are not based on educational justifications. Play without adult interference can allow children to follow their interests; this is important considering an understanding of children as competent human beings [81] and their autonomous learning and development [66]. However, it is suggested that teachers should actively involve themselves in children's play to expand the potential learning and development [17] and to be able to meet children's emotional needs [20].

Based on the results, it is difficult to assess to what extent teachers observe and intervene in children's free and unstructured play. However, another prominent type of teacher support of play is denoting children in playful situations with teachers, and play situations where teachers take participating actions. In other words, teachers interact intentionally in children's play. In the synthesis, playful situations were called 'playful reasoning' and 'play-based learning'. These situations were described as communicative in a verbal sense, where teachers promoted positive experiences and playful interactions. Through direct communication, teachers potentially contribute to expanding children's current developmental stage [34]. Communication was often based on the children's activities or interests and related to objects in the natural environment. Teachers helped children obtain knowledge, mainly in playful and instructional ways. Thus, the question arises whether these activities can be considered as play. It might be argued that playful situations reflect a learning rather than a play concept. Nevertheless, play and learning are often considered inseparable in ECEC practice [14]. The synthesis provides a varied picture of teachers' approaches to playful situations. Several of these were child-centered, in the sense of being child-initiated or child-led. Other approaches were described as more teacher-guided, teacher-led, and teacher-directed. These types of teacher support can be interpreted as providing learning experiences in playful contexts, where the teacher has educational intentions for children's learning and development. Playful situations may also function as a facilitating external condition for play, thus forming a symbiotic relationship between different types of teacher support. When interpreting these findings, one may also focus on teachers' individual preferences and preconceptions, i.e., that different teachers have different ways of supporting play in nature-based outdoor spaces, as suggested in the synthesis.

Most teachers practice encouraging and facilitating actions addressing the children and the play environment or interacting in playful situations with children. It can be assumed that many teachers carry out both. Nevertheless, the included papers briefly described situations where teachers participate in children's free and symbolic play. The reasons why this type of teacher support has low visibility in the synthesis may lie in the differences related to the types of studies, with varying research questions and aims. Primarily, an obvious explanation could be that none of the articles explored our phenomenon of interest explicitly. On the other hand, the phenomenon that seldom occurs might be a consequence of that teacher acting in forests and woodlands, possibly placing the most significance on the inherent properties of the natural environment itself. Yet, it can be assumed that teachers participating in children's free and symbolic play can incorporate children's independent mobility license [21] and children's learning and development

with the support of an adult in situ [34]. The presence of an adult in peer interactions can provide opportunities for teachers to meet children's emotional needs [20]. As we see it, teachers must facilitate external conditions based on knowledge of the relation between the potential of the natural environment and the children's play. At the same time, teachers must be sensitive to their interaction and participation in play to meet the children's needs of autonomy, development, learning, and social and emotional support.

## 5. Limitations

Within the limitations of this review, the conceptualization of play and teachers' support of play raises some methodological questions. There are different ways of understanding the concepts, and, therefore, our conceptualizations can be challenged too. The authors' professional and theoretical background have influenced the search string's design as well as how we read and interpret the papers in the screening processes, data extraction, and synthesis. The inclusion and exclusion criteria may have caused us to miss important papers: for example, publications written in other languages, publications that might apply other terms than applied by us, and other types of publications than empirical peer-reviewed journal articles. The choice of databases may also have restricted possible hits.

In the initial phase, the authors developed a protocol for a scoping review [82] to provide an overview of the research literature. However, in dialogue between the authors and the emerging literature, we found it more relevant to synthesize the material. Hence, the method for conducting this review turned towards a 'traditional' systematic review. Regarding synthesizing, we applied core elements of Noblit and Hare's meta-ethnography method [44]. This method was adopted underway; hence, it was not used strictly. For example, we did not adopt themes based on reciprocal translation but narratively incorporated perspectives of teacher support; this can be seen as inconsistent in methodological stringency and may have influenced the results.

## 6. Implications

This review provides a foundation to argue that, in addition to understanding the educational potential of children's encounters with natural environments, there is a need for researchers and teachers to articulate the educational justifications and reasons for the type of teachers' support given to the children. Studies that can further examine teachers' attitudes, beliefs, and intentions when it comes to supporting children, would complement the findings of this article. In the included papers, there are few conceptualizations of the terms 'child-centered' and 'children's interests', although these are prominent concepts in several studies. Increased knowledge of these concepts can benefit the understanding of teacher support. This also applies to studies examining children's voices by analyzing how children experience teacher support and perceive their play in rugged terrains and with natural materials. The results suggest that the training and experience of teachers may influence the type of support and how support is given to the children, which should receive more attention in ECEC teacher education and professional development. The results reveal that the traditional child–adult dichotomy seems to be challenged in natural environments. This could be a starting point in teachers' reflection. As we see it, this finding provides a promising base for a more child-centered educational practice, which can benefit play and, by that, children's well-being, learning and development.

In the discussion we emphasized the low visibility and attendance of teachers taking participating actions in children's free and symbolic play as particularly interesting. This lack of research-based knowledge needs to be addressed to explore how teachers participate in this type of play, why they do it, and if they do not, why they refrain from doing it.

**Supplementary Materials:** The following supporting information can be downloaded at: https://www.mdpi.com/article/10.3390/educsci14010013/s1. File S1: PRISMA checklist; File S2: Search string; File S3: Included articles; File S4: Table and quality appraisal.

**Author Contributions:** Conceptualization, T.M.S., R.K., R.L. and T.M.; methodology, T.M.S., R.K., R.L. and T.M; validation, T.M.S., R.K., R.L. and T.M; writing—original draft preparation, T.M.S.; writing—review and editing; T.M.S., R.K., R.L. and T.M; formal analysis, T.M.S., R.K., R.L. and T.M; data curation, T.M.S.; supervision, R.K., R.L. and T.M.; visualization, T.M.S.; project administration, T.M.S. All authors have read and agreed to the published version of the manuscript.

**Funding:** This research was funded as part of a Public Sector PhD project by the Research Council of Norway, project number 333244, and Kanvas Foundation, Norwegian organization number 971 272 643.

**Institutional Review Board Statement:** Not applicable.

**Informed Consent Statement:** Not applicable.

**Data Availability Statement:** Data are contained within the article.

**Conflicts of Interest:** The authors declare no conflict of interest.

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
