# Peer review of "Early Childhood Teachers’ Support of Children’s Play in Nature-Based Outdoor Spaces—A Systematic Review"

_education, doi:10.3390/educsci14010013_

Round 1
Reviewer 1 Report
Comments and Suggestions for Authors
This is an interesting paper that has potential. I feel there are a few overarching issues that, if resolved, would result in a more valuable paper:
1. The decision to weave the discussion as a narrative rather than presenting teachers’ strategies for supporting children’s play in a thematic approach has resulted in a lack of clarity about the types of strategies used. I would have preferred to see the findings/discussion presented with greater clarity.
2. The discussion would benefit from greater links to the theory discussed in the introduction.
3. Amending the discussion of reported findings to past tense throughout might read more appropriately for a systematic review.
|
Page 1, line 19 - 21 |
Punctuation/wording could be revised for easier reading |
|
Page 1, line 26-27 |
‘has given value to’? interesting wording – could be revised for greater clarity |
|
Page 1, line 29-32 |
Rewording may aid clarity of concepts here |
|
Page 1, line 33 |
What do you mean by ‘group processes’? |
|
Page 1, line 35 Page 2, line 48/63 etc. |
Use of the term ‘institutions’ doesn’t sit well with me – ‘settings’ is a more appropriate term |
|
Page 2, line 84, |
Need – needed (past tense) |
|
Page 5, line 202-203 |
I’m not sure this sentence makes sense? Might need rewording. |
|
Page 7, section 3.4 |
Clearer articulation/expression of the themes of teachers support e.g. perhaps through the use of subheadings. I feel the themes get a bit lost in the narrative |
|
Page 7, line 292-293 |
Including some punctuation here would help to clearly identify the phrases used to describe the main features of teachers support of play e.g. t is not clear what is meant be ‘the actual situation’ |
|
Page 8, line 311-312 |
‘bears in the woods’ comment needs further explanation – what did the teacher mean by this? And how did this encourage/lead to children’s play. Detail here is important given the focus of the paper. |
|
Page 8, line 314 |
‘teachers were sensitive to their influence’ – their own influence? And on who? Clarity here would be good. |
|
Page 8, line 320-322 |
This sentence needs revising for clarity. |
|
Page 4, line 323 |
Offer - offers |
|
Page 8, line 326-327 |
Encouraging and withdrawing from play are 2 very different concepts. It might be clearer if you discuss these separately. Are they the same papers that talk about both? |
|
Page 8, line 327 |
‘it is observed’? wording doesn’t seem right for this context as you are reporting on what has been observed and reported by others rather than observing it directly. The choice of ‘it is’ also might need amending to past tense. |
|
Page 8, line 351 |
Reminding the reader of the theory discussed in the introduction would be useful. |
|
Page 8, line 353-354 |
First sentence ‘of the provision’ – not sure what this means? |
|
Page 9, line 350 |
You indicate that the discussion will follow as per the results, yet you have combined aims and methods with the core concepts – this suggests you didn’t have much to say about either. It would be good to see them both get significant attention (otherwise why report on them?). I feel that the core concepts of play and teachers support of play (included in the discussion in 4.2) has been overlooked. |
|
Page 9, line 374-375 |
The first sentence seems out of place in this section. The second sentence is a bit unclear |
|
Page 9, line 378-381 |
More detail here might help to understand your explanation for the differences in findings e.g. what kind of research led to Hindmarch and Boyds findings? Were they looking at curricula or was it empirical research? You identify that different methods can bring different results – indicating that noting the different methods might be important for comparison. |
|
Page 9, line 389-391 |
More detail here to identify the significance of this comment would be useful. E.g. what were the exceptions regarding bush-kinder settings in Australia and what does this tell us? |
|
Page 9, line 398-399 |
This last sentence of this section is a bit confusing. |
|
Page 10, line 413-415 |
The most prominent type….includes: ‘the’ doesn’t correlate with ‘includes’
Not sure about the use of tenses here e.g. ‘teachers who are facilitating’. Past tense to refer back to the reviewed papers might be better.
Greater clarity in the results section (see comment above re page 7, line 292) would help to link here. I am a bit confused about:
‘facilitating actions addressing the physical environment’ (p.7) and ‘facilitating desirable actions and processes’ (p.10)
I think you are describing the same thing in different ways, however what you mean in both cases is not clear to me.
The detail further down (line 417-419) is useful and provides more clarity.
|
|
Page 10, line 14 |
A brief explanation of Prins, Van Der Wilt, Van Der Veen, and Hovinga’s conceptualisation of play quality would be useful. |
|
Page 10, line 434 |
Relation - relationship |
|
Page 10, line 435 |
Tense
Appeared? Taken place? |
|
Page 11, line 468 |
The continuum sentence might be useful at the beginning of the discussion |
|
Page 11, line 494-496 |
Here you discuss the conceptualisations of concepts – even though this was left out of the discussion (see previous note) – correlating these sections would be helpful. |
|
Page 12, line 525-528 |
I can see you have identified your choice incorporate findings in narrative rather than through themes – as per my previous comment I feel a thematic approach might have provided more clarity. |
n/a
Reviewer 2 Report
Comments and Suggestions for Authors
Thank you for the opportunity to read and review this paper, which I enjoyed and did with interest. This was a well-done review that makes a meaningful contribution to the literature, particularly with regards to play, outdoor play, and education more broadly. The methodology, results, and discussion were all appropriate (and I commend the work involved in this). However, I do think the inclusion of additional context to the introduction, more clarity rationale in some parts of the methods section, and some other minor additions is needed before acceptance. Please see below for specific comments:
Introduction
1. This section is two paragraphs, the first of which contains a lot of information that should be separated into relevant paragraphs with more information included (see below). From my reading, there are several key points that would be their own paragraphs:
a. Introduction to education in outdoor environments and their benefits.
b. Introduction to play and play in natural environments and their benefits
c. Introduction to the role of teachers in supporting group processes including education and play in outdoor environments
d. Introduction to the study.
2. There were several points in the introduction that would benefit from more information:
a. Lines 16-17 – What aspects of children’s health, learning and development is being in natural environments beneficial for?
b. Lines 17-18 – Why is using natural environments in ECEC of interest to authorities, educators and researchers?
c. Lines 50-60 – What is known about children’s play in outdoor spaces more generally? The introduction gets straight to teachers’ support of outdoor play, but there’s no summary of what is known about children’s outdoor play more broadly before getting into that.
Methods
3. A justification for the time of publication is needed – why was this restricted to studies published from 2002 (line 72)?
4. It would be helpful to include a justification/rationale for the choice of databases for this topic (lines 90-96).
5. For efficiency, the information in Tables 1 and 2 could be embedded within the respective boxes in Figure 1.
Results
6. Given the results speak to different kinds of outdoor environments (lines 192-224) and different kinds of play (lines 257-260), these should be introduced and discussed in the introduction (e.g., what are these different kinds of play types and why is the distinction between them important?).
7. I would advise including a sentence or two in the results section that briefly summarises the ‘take home’ message of Appendix D – from my reading, all articles are rated medium-high, which is great and I think could be briefly stated either at the beginning or end of the introduction for the benefit of a reader who might miss the appendix.
Discussion
8. I appreciate the discussion of the difference between studies from the Nordic and Anglosphere countries (lines 361-388), but it would also be of benefit to discuss the missingness of studies from outside the Global North and the implications of this.
9. I would suggest swapping sections 5 and 6 in terms of order – that way the paper ends on the implications and ‘take-home’ messages of this nice piece of work.
Round 2
Reviewer 2 Report
Comments and Suggestions for Authors
Thank you for the opportunity to read this revised version of the manuscript. The added content, in my opinion, has improved the overall quality of the work.
Comments on the Quality of English LanguageI think the paper would benefit from a careful proof-read at the proofing stage.